# Prognosis of Pancreatic Cancer Based on Resectability: A Single Center Experience

**DOI:** 10.3390/cancers15041101

**Published:** 2023-02-09

**Authors:** Takahiro Einama, Yasuhiro Takihata, Suefumi Aosasa, Fukumi Konno, Kazuki Kobayashi, Naoto Yonamine, Ibuki Fujinuma, Takazumi Tsunenari, Akiko Nakazawa, Eiji Shinto, Hideki Ueno, Yoji Kishi

**Affiliations:** 1Department of Surgery, National Defense Medical College, Saitama 359-8513, Japan; 2Departmetn of Surgery, Shinkuki General Hospital, Sasitama 346-0021, Japan

**Keywords:** pancreatic cancer, resectability, conversion surgery

## Abstract

**Simple Summary:**

Conversion surgery has been increasingly performed for initially unresectable advanced pancreatic ductal adenocarcinoma (PDAC). Patients with PDAC were classified into three groups: resectable (R), borderline resectable (BR), and unresectable (UR). In total, 211 patients (R, 118; BR, 22; UR, 81) were selected. Among them, 117 (99%), 18 (82%), and 15 (19%) patients in the R, BR, and UR groups underwent surgical resection. R0 resection rates were 88%, 78%, and 67%, whereas median overall survival (OS) rates from treatment initiation were 31, 18, and 11 months (*p* < 0.0001) in the R, BR, and UR groups, respectively. In patients who underwent surgical resection, relapse-free survival (RFS) and OS were similar. Lymph node metastases and incomplete adjuvant chemotherapy were identified as independent prognostic factors for OS.

**Abstract:**

Although conversion surgery has increasingly been performed for initially unresectable advanced pancreatic ductal adenocarcinoma (PDAC), the rate of conversion, including that for patients who do not undergo resection, remains unclear. Patients with PDAC who were treated between January 2013 and December 2018 were classified into three groups: resectable (R), borderline resectable (BR), and unresectable (UR). We analyzed patient outcomes, including the rate of surgical resection and survival, in each of these groups. In total, 211 patients (R, 118; BR, 22; UR, 81) were selected. Among them, 117 (99%), 18 (82%), and 15 (19%) patients in the R, BR, and UR groups, respectively, underwent surgical resection. R0 resection rates were 88, 78, and 67%, whereas median overall survival (OS) from treatment initiation were 31, 18, and 11 months (*p* < 0.0001) in the R, BR, and UR groups, respectively. In patients who underwent surgical resection, relapse-free survival (RFS) and OS were similar among the three groups (R vs. BR vs. UR; median RFS (months), 17 vs. 13 vs. 11, *p* = 0.249; median OS (months), 31 vs. 26 vs. 32, *p* = 0.742). Lymph node metastases and incomplete adjuvant chemotherapy were identified as independent prognostic factors for OS. Although the surgical resection rate was low, particularly in the BR and UR groups, the prognosis of patients who underwent surgical resection was similar irrespective of the initial resectability status.

## 1. Introduction

Pancreatic ductal carcinoma (PDAC) remains one of the most lethal malignancies. Despite the development of strategies to detect and manage pancreatic cancer with respect to surgical techniques [1,2], postoperative management [3], neoadjuvant chemotherapy [4,5], and adjuvant chemotherapy [6], only approximately 4% of patients remain alive for 5 years after diagnosis of PDAC [7].

To improve the prognosis of patients, we created a multidisciplinary treatment team that includes gastroenterologists, radiologists, surgeons, and pathologists. Treatment plans are selected according to the resectability status, which is classified into resectable (R), borderline resectable (BR), and unresectable (UR), according to the National Comprehensive Cancer Network (NCCN) *Clinical Practice Guidelines in Oncology*, *Pancreatic Adenocarcinoma*, version 2. 2017 [8]. Once resectability status was decided at the team conference, we performed upfront surgery and adjuvant chemotherapy for R patients [6,9] and chemotherapy for BR and UR patients in each department, as the standard care protocol. Our team used these treatment strategies for patients with pancreatic cancer until preoperative chemotherapy was applied to patients with R patients [10].

In recent years, we have increasingly performed conversion surgery with the emergence of more effective chemotherapy regimens, such as FOLFIRINOX (fluorouracil, leucovorin, irinotecan, and oxaliplatin) [11], gemcitabine plus nab-paclitaxel [12], and nanoliposomal irinotecan with fluorouracil and folinic acid [13]. However, few studies have examined the conversion rates of BR and UR patients or the clinical outcomes of pancreatic cancer patients, including those who failed to undergo surgery [14,15,16,17,18,19].

Therefore, we herein conducted a retrospective analysis of the clinical course of PDAC patients in each resectability status group, with or without surgery, and attempted to identify clinical issues with the goal of prolonging survival.

## 2. Materials and Methods

The present study was approved by the Institutional Review Board of the National Defense Medical College (Approval No. 3038). All participants provided informed consent.

### 2.1. Patients

Patients diagnosed with PDAC between January 2013 and December 2018 at the National Defense Medical College were selected. All patients had cytologically or pathologically proven ductal carcinoma of the pancreas, and their cases were discussed with respect to their treatment plans at a multidisciplinary treatment team meeting that includes gastroenterologists, radiologists, surgeons, and pathologists. Each patient was classified into the following groups based on the resectability status defined by NCCN: R, BR, and UR disease. BR disease was further classified into BR-A, in which the tumor contacted <180° of the celiac, hepatic, or superior mesenteric artery, and BR-PV, in which the tumor contacted ≥180° of the portal or superior mesenteric vein but did not contact the aforementioned arteries. UR disease was classified into UR-LA, in which tumor contact with adjacent vascular structures was beyond the criteria of BR, and UR-M, in which distant metastases were recognized [8].

Our treatment strategy during the study period was as follows. In patients with R, we performed upfront surgery followed by adjuvant chemotherapy with S-1 for 6 months. In patients with BR and UR, systemic chemotherapy was the first-line treatment. After chemotherapy for at least several months, surgical resection was considered based on CT image findings. We discussed and decided on the indication for conversion surgery for individual UR patients at a multidisciplinary treatment team meeting. Conversion surgery was permitted for only those who met the following conditions: patients showing adequate reduction of the main tumor, enabling complete removal inclusive of the major vessels and metastatic site; those with no metastasis; or those with controllable metastasis by surgical resection. We performed adjuvant chemotherapy with S-1 for 6 months after pancreatectomy. The postoperative chemotherapy regimen used for each patient was S-1 monotherapy based on the JASPAC 01 trial [6]. The regimen consisted of S-1 80–120 mg/ day, according to body surface area (<1.25 m^2^: 80 mg/day; 1.25–1.50 m^2^: 100 mg/day; >1.50 m^2^: 120 mg/day), which was orally administered twice a day for 28 days followed by 14 days rest and repeated every 42 days (1 cycle) for 4 courses. Alternatively, treatment was orally administered twice a day for 14 days followed by 7 days rest and repeated every 21 days (1 cycle) for 8 cycles.No patients received perioperative radiotherapy [20]. Postoperative surveillance was performed through examinations of tumor markers every 3 months and CT every 6 months. PET/CT was also conducted to detect recurrence. R0 was defined as pathologically margin free in the resected specimen.

### 2.2. Statistical Analysis

Survival curves were generated by the Kaplan–Meier method and compared using the log-rank test. Overall survival was calculated as the interval between initial treatment and death due to any cause (including death from other diseases), whereas relapse-free-survival was calculated as the interval between surgery and disease progression. Univariate and multivariate analyses of predictors for worse relapse-free survival (RFS) and overall survival (OS) were performed using the Cox proportional hazard model, and outcomes are shown with the hazard ratio (HR) and 95% confidence interval (CI). Statistical analyses were performed using JMP^®^ 14 (SAS Institute Inc., Cary, NC, USA).

## 3. Results

### 3.1. Resection Rate/Portal Vein Resection and Arterial Resection Rate/Residual Tumor Rate According to the Resectability Status

In total, 211 patients were examined: 118 in the R group, 22 in the BR group, and 81 in the UR group. In the R group, we performed upfront curative-intent resection on 117 patients (99%). One patient did not undergo resection due to respiratory dysfunction. The BR group included 5 patients with BR-PV and 17 patients with BR-A. We performed upfront surgery on two patients with BR-PV and three with BR-A. Surgical resection following systemic chemotherapy was performed on 2 out of 3 patients (67%) in the BR-PV group and 11 out of 14 patients (79%) in the BR-A group. The total resection rate in the BR group was 82% (18/22). The UR group included 38 patients with UR-LA and 43 with UR-M. We performed upfront surgery on two patients with UR-LA disease. The conversion surgery rate was 28% (10/36) in the UR-LA group and 7% (3/43) in the UR-M group, in which the first patient had lung metastasis and received gemcitabine plus erlotinib hydrochloride for 6 months, the second patient had liver metastasis and received FOLFIRINOX (a combination of leucovorin, fluorouracil, irinotecan, and oxaliplatin) for eight cycles, and the third patient had para-aortic lymph node metastasis and received gemcitabine plus nab-paclitaxel for six cycles. The total resection rate in the UR group was 19% (15/81). The total resection rate of PDAC between January 2013 and December 2018 was 71% (150 out of 211 patients).

We performed portal/superior mesenteric vein resection, hepatic/superior mesenteric arterial resection, and both portal vein and arterial resection on 21 (18%), 2 (2%), and 1 patient (1%), respectively, in the R group; 13 (72%), 0, and 1 patient (6%), respectively, in the BR group; and 1 (7%), 0, and 8 patients (53%), respectively, in the UR group. 

R0 resection rates were 88.0% (103 of 117) in the R group and 78% in the BR group (for the BR group, 14 out of 18 (78%); for the BR-PV group, 4 out of 4 (100%); for the BR-A group, 10 out of 14 (71%)). R0 resection was performed on only one out of three patients (33%) by upfront surgery in the BR-A group. In the UR group, the R0 resection rate was 67% (in the UR group, 10 out of 15 (67%); in the UR-LA group, 10 out of 12 (83%); in the UR-M group, 0 out of 3 (0%)) (Figure 1).

Mortality occurred in two patients (1%): an 82-year-old male with R-PDAC who died of stroke on postoperative day 11 after subtotal stomach-preserving pancreaticoduodenectomy (SSPPD) and a 73-year-old male who died of pneumonia on postoperative day 154 after SSPPD.

### 3.2. Total Survival, RFS, and OS of Resection Cases

Median follow-ups were 29.5 months (range of 0.4–89.0 months) in the R group, 24.5 months (range of 3.6–80.9 months) in the BR group, and 14.5 months (range of 0.7–59.0 months) in the UR group. Total median OS in the R, BR, and UR groups were 31.4, 18.1, and 11.2 months, respectively. In the BR group, the median OS of patients with and without surgery were 26.2 and 9.3 months, respectively. In the UR groupthe median OS of patients with and without surgery were 32.1 and 7.3 months, respectively (Figure 2). In patients who underwent surgical resection, RFS and OS rates were both similar among the three groups. However, 3-year RFS was poorer in the BR (16.5 months) and UR (19.1 months) groups than in the R group (33.2 months) (Figure 3).

### 3.3. Predictive Factors for RFS and OS in Patients Who Underwent Surgery

Among patients who underwent surgical resection, the pathological T-factor, lymph node metastases (N+), lymphatic permeation, residual tumor, and no or incomplete adjuvant chemotherapy were identified as significant risk factors for worse RFS in a univariate analysis. A multivariate analysis of these parameters showed that N+ (*p* = 0.0002; HR = 3.96; 95% CI, 1.83–10.00) and no or incomplete adjuvant chemotherapy (*p* < 0.0001; HR = 2.82; 95% CI, 1.81–4.46) were independent predictors of worse RFS (Table 1). Regarding OS, N+, lymphatic permeation, residual tumor, and no or incomplete adjuvant chemotherapy were significant predictors of worse OS in a univariate analysis. A multivariate analysis of these parameters showed that N+ (*p* = 0.0012; HR = 3.24; 95% CI, 1.54–7.79) and no or incomplete adjuvant chemotherapy (*p* < 0.0001; HR = 2.88; 95% CI, 1.81–4.66) were independent predictors of worse OS (Table 2).

## 4. Discussion

We herein present the outcomes of PDAC patients based on their resectability status. Conversion surgery prolonged survival in the BR and UR groups. Furthermore, we identified N+ and incomplete adjuvant chemotherapy as independent prognostic factors in patients who underwent surgical resection. The majority of studies that examined the outcomes of conversion surgery for PDAC focused on locally advanced cases, including BR and UR-LA disease [16,21] or either [14,22,23]. There has been no study that comprehensively showed survival according to the initial resectability status, including patients who did not undergo surgical resection, in a single institution.

To achieve better long-term outcomes, the present results suggest that it is important to complete adjuvant chemotherapy. The JASPAC 01 trial compared OS with and without postoperative adjuvant chemotherapy with S-1 after pancreatectomy and showed that 28% of patients in the S-1 group discontinued treatment before completion [6]. The main reasons for discontinuation were decisions by physicians or refusal due to adverse events in 21% followed by tumor recurrence in 5%. Iwasa et al. recommended skipping the administration of S-1 if patients developed gastrointestinal toxicities and reinitiating S-1 again after the attenuation of symptoms [24]. Kobayashi et al. reported that maintaining a total dose intensity of at least 60% in S-1 adjuvant chemotherapy seems important to achieve a long postoperative survival in patients with pancreatic cancer [20]. To complete the planned 6-month treatment, we need appropriate guidelines that are established for the proper management of adjuvant S-1 therapy.

In patients who underwent surgical resection, the initial resectability status was not identified as a significant risk factor for worse RFS or OS in the univariate or multivariate analysis. The anatomical definition of BR is a tumor that is at high risk for R1 resection, particularly due to tumor exposure around vascular structures when surgery is used as an initial treatment strategy [25]. Neoadjuvant chemotherapy and/or radiotherapy is considered to increase the probability of R0 resection [26]. 

We have performed R0 resections in 78% of BR patients who underwent pancreatectomy, this result proved this suggestion. 

This value was close to 88% in the R group (88%) irrespective of the high frequency of portal vein/superior mesenteric vein resection (PVR) of 72%. Yamaguchi et al. showed that neoadjuvant chemotherapy for BR patients was well tolerated, with a conversion surgery rate of 84% and an R0 resection of 67%. Furthermore, intention-to-treat analysis showed favorable 3-year overall survival rate of 54.7% and median survival rate of 39.4% [23]. Based on these findings, we may provide chemotherapy as a first-line treatment and need to consider conversion surgery to prolong survival during chemotherapy for BR patients. 

The prognosis of patients in the UR group was dismal. Conversion rates were low at 28% (10 out of 36) in the UR-LA group and 7% (3 out of 43) in UR-M group. To achieve R0 resection, particularly for initial UR-LA disease, we need to perform extensive surgical resection, including PVR and/or arterial resection. Regarding the resection area of the SMA plexus for R0 resection, we decided the resection area based on CT before chemotherapy [27]. In our experience of conversion surgery in the UR group, PVR and arterial resection were performed on eight patients (53%). The R0 rate was 67% (10 of 15), and postoperative mortality occurred in one patient. The feasibility of combined arterial resection, particularly in terms of long-term survival, is controversial. Regarding survival, Kimura et al. showed that conversion surgery in UR patients may prolong survival; however, conversion rates were low at 18% (12 of 66) in UR, 24% (10 of 42) in UR-LA, and 8% (2 of 24) in UR-M [14]. Furthermore, there was one case of hepatic artery resection (8%) and no mortality. Yanagimoto et al. also suggested that conversion surgery offered benefits in terms of prolonged survival for initial UR patients who responded favorably to chemotherapy when combined with postoperative adjuvant chemotherapy; there were three cases of common hepatic artery resection (9%) and four of celiac artery resection (13%) with no mortality [22]. Bichenbach et al. reported no significant difference in OS from the time of resection between UR-LA and R patients. Regarding complications [28], Gemenetzis et al. showed that the morbidity rate was 59% (Clavien–Dindo classification higher than III), and the 90-day mortality rate was 4%. Morbidity and mortality rates were not higher than those in R cases [21]. Therefore, if the surgical technique of arterial resection is established, it is also important not to miss the opportunity for surgical resection in UR cases during chemotherapy because RFS and OS were better in the UR group with surgery than in the UR group without surgery.

Preoperative chemotherapy offers several theoretical advantages over upfront surgery, including the early delivery of systemic chemotherapy, the high tolerance of multi-agent regimens by patients, and a higher negative-margin resection rate, and is thought to prolong OS. Resection status, which other research has demonstrated to be an important independent predictor of survival [15,23], was not revealed as a prognostic factor in our study. This might be the influence of favorable effect of adjuvant chemotherapy or inclusion of patients with initially unresectable diseases. Furthermore, UR patients with surgery are basically considered to be a group of patients who responded well to treatment with chemotherapy, and even if recurrence occurs, many of them might survive for a long period of time due to good response to subsequent chemotherapy. We performed chemotherapy as a first-line treatment for the BR and UR groups. This may allow for the better selection of patients for whom chemotherapy may not be effective because of high malignant potentiality, an increased probability of margin negative resection, and the treatment of micrometastases [29]. The optimal regimen and duration for chemotherapy has not yet been established. Combination regimens, such as FOLFIRINOX or gemcitabine plus nab-paclitaxel, are generally preferred for patients with a preserved performance status and no relevant comorbidities due to positive results observed in the metastatic setting [30].

The limitations of the present study include its retrospective nature and small sample number. The retrospective cohort included 211 patients, which is considerable for PDAC. However, 43 patients, that is, 20% of the study population, had metastatic disease. It is very likely that the impact of surgery on the survival of patients with systemic disease is very low if not null. Therefore, it is expected that their inclusion in analyses negatively affects survival estimations. Further studies with larger numbers of patients are needed to validate the present results and show the usefulness of conversion surgery in the BR and UR groups. Furthermore, the effects of neoadjuvant chemotherapy in the R group were not evaluated because neoadjuvant chemotherapy was initiated only after 2019, following the findings of PREP-02/JSAP05 [10], which showed prolonged survival in patients who received preoperative gemcitabine plus S-1 treatment compared to those with upfront surgery for resectable or BR-PV disease. However, there has been no study that comprehensively showed survival according to the initial resectability status or whether surgical resection was performed in a single institution.

## 5. Conclusions

We need to consider conversion surgery in the BR and UR groups during chemotherapy because of the better prognosis observed in the R group. Furthermore, as completion adjuvant chemotherapy was an independent prognostic factor in surgery cases, we also need to ensure the proper management of adjuvant S-1 therapy in order to complete the planned 6-month treatment.

## Figures and Tables

**Figure 1 cancers-15-01101-f001:**
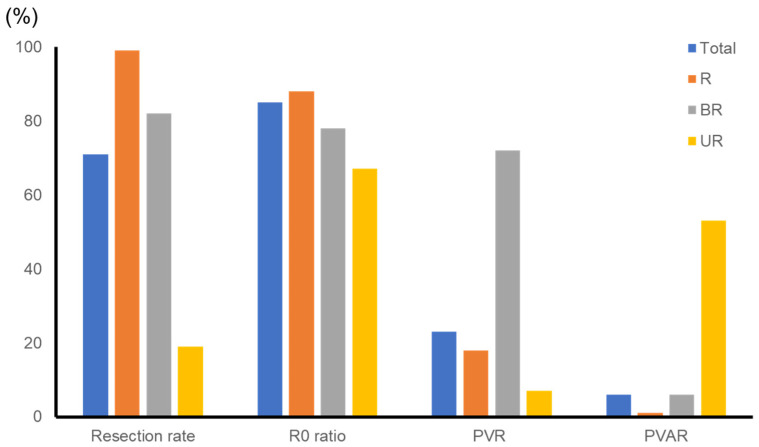
Total resection rate and status and combined vascular resection. R: resectable, BR: borderline resectable, UR: unresectable, PVR: portal/superior mesenteric vein resection, PVAR: both portal/superior mesenteric vein and arterial resection.

**Figure 2 cancers-15-01101-f002:**
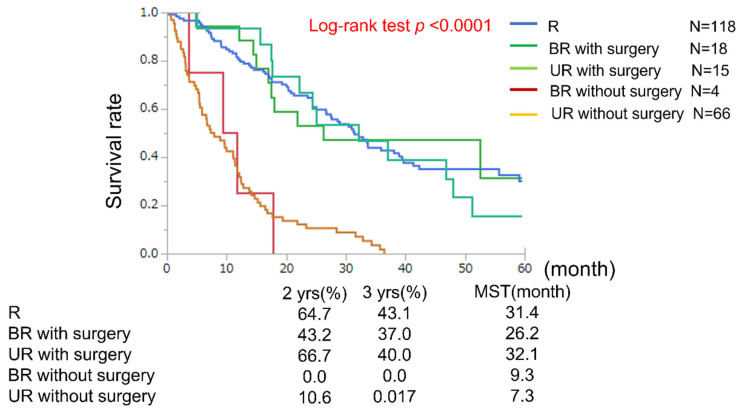
Overall survival classified by resectability in resection cases. The curve for the patient groups in which R—resectable, BR—borderline resectable with surgery, and UR—unresectable with surgery significantly differed from that for the BR and UR groups without surgery (*p* < 0.0001).

**Figure 3 cancers-15-01101-f003:**
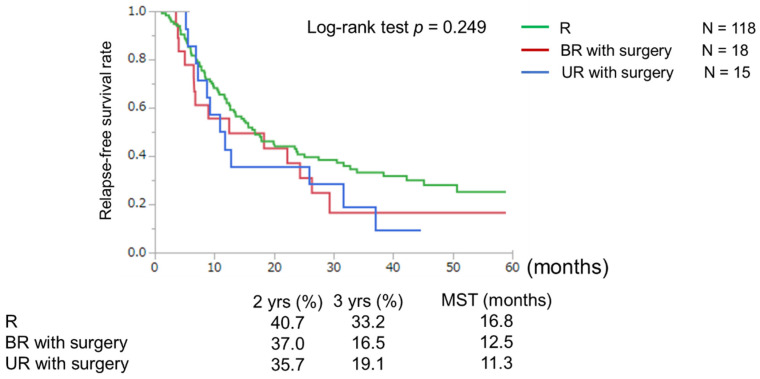
Relapse-free-survival classified by resectability in resection cases. The curve for the patient group in which R—resectable was not significantly different from that for the patient groups BR—borderline resectable with surgery and UR—unresectable with surgery (*p* = 0.249).

**Table 1 cancers-15-01101-t001:** Predictive factors for relapse in surgical cases.

	Univariate		Multivariate	
Parameter (Unfavorable vs. Favorable)	HR (95% CI)	*p*-Value	HR (95% CI)	*p*-Value
Pathological T-factor	3.61	0.0248	2.87	0.236
(T3,4 vs. T1,2)	(1.14–87.4)		(0.57–52.16)	
Pathological N-factor	3.53	<0.0001	3.959	0.0002
(Positive vs. Negative)	(1.87–7.56)		(1.83–10.00)	
Lymphatic permeation	2.99	0.0011	1.14	0.767
(Positive vs. Negative)	(1.49–7.12)		(0.44–2.50)	
Residual tumor	1.98	0.0194	1.52	0.156
(R1 > R0)	(1.12–3.29)		(0.84–2.61)	
Completed adjuvant chemotherapy	2.47	<0.0001	2.82	<0.0001
(No > Yes)	(1.62–3.80)		(1.81–4.46)	
Resectability	0.1088			
(BR + UR vs. R)	(0.44–1.09)			

HR: Hazard ratio, R: resectable, BR: borderline resectable, UR: unresectable.

**Table 2 cancers-15-01101-t002:** Predictive factors for overall survival in surgical cases.

	Univariate		Multivariate	
Parameter (Unfavorable vs. Favorable)	HR (95% CI)	*p*-Value	HR (95% CI)	*p*-Value
Pathological T-factor	2.68	0.168		
(T3,4 vs. T1,2)	(0.85–16.30)			
Pathological N-factor	2.69	0.0010	3.24	0.0012
(Positive vs. Negative)	(1.44–5.57)		(1.54–7.79)	
Lymphatic permeation	2.43	0.0106	1.029	0.948
(Positive vs. Negative)	(1.21–5.81)		(0.409–2.23)	
Residual tumor	1.96	0.0147	1.69	0.0724
(R1 > R0)	(1.15–3.18)		(0.95–2.87)	
Completed adjuvant chemotherapy	2.54	<0.0001	2.88	<0.0001
(No > Yes)	(1.62–4.07)		(1.81–4.66)	
Resectability	1.12	0.622		
(BR + UR vs. R)	(0.693–1.761)			

HR: Hazard ratio, R: resectable, BR: borderline resectable, UR: unresectable.

## Data Availability

All data generated and analyzed during this study can be retrieved by sending a formal request by email to the corresponding author.

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
