# Peer review of "Prognosis of Pancreatic Cancer Based on Resectability: A Single Center Experience"

_cancers, 2023, doi:10.3390/cancers15041101_

Round 1

Reviewer 1 Report

This paper is well written. As a retrospective study, this is modestly useful information. There is no mention of the role of radiotherapy in this setting. I would view this paper as acceptable for publication

Author Response

Comment 1:

This paper is well written. As a retrospective study, this is modestly useful information. There is no mention of the role of radiotherapy in this setting. I would view this paper as acceptable for publication.

Response 1;

Thank you for your comment, and I am very happy to know your recommendation for publication.

In this series, we didn’t perform chemoradiotherapy in any patient including those with BR and UR-LA diseases. We added “No patient received preoperative radiotherapy” in Line 80.

Reviewer 2 Report

Authors present a retrospective study evaluating prognosis of PDAC based on resectability.

The paper is well written and the topic is of general interest for the community of oncologists.

I have few observations.

1)      The study, although relatively well designed and conducted, suffers a certain lack of originality and novelty. Indeed, many studies have been published dealing with this issue, and several meta-analyses have been provided. Consequently, the concept that resectability, either upfront or obtained after neoadjuvant therapy, positively affects survival in PDAC is now widely accepted. Moreover, observed survival and conversion rates largely reflects what reported by other studies.

2)      Lines 54-56: “few studies have examined the conversion rates of BR and UR patients or the clinical outcomes of pancreatic cancer patients, including those who failed to undergo surgery”, references are laking. The sentence needs to cite references for these studies.

3)      The retrospective cohort includes 211 patients, which is considerable for PDAC. However, 43 patients (20% of the study population) have metastatic disease. It is very likely that the impact of surgery on survival of patients with systemic disease is very low if not null. Therefore, it is expected that their inclusion in analyses negatively affects survival estimations.

4)      While termination of adjuvant chemotherapy and N+ status were confirmed as prognostic factors, surprisingly enough the presence of residual tumor after surgery failed to predict survival in multivariate analysis. Indeed, several studies have demonstrated that resection margins (R) status is an important independent predictor of survival: its lack of significance in Cox analysis rises to concerns about criteria used for patient selection.

5)      Lines 177-178 and 182-183, the following sentence appears repeated: “To complete the planned 6-months treatment, we need appropriate guidelines that are established for the proper management of adjuvant S-1 therapy.”

Author Response

Comment 1:

The study, although relatively well designed and conducted, suffers a certain lack of originality and novelty. Indeed, many studies have been published dealing with this issue, and several meta-analyses have been provided. Consequently, the concept that resectability, either upfront or obtained after neoadjuvant therapy, positively affects survival in PDAC is now widely accepted. Moreover, observed survival and conversion rates largely reflects what reported by other studies.

Thank you for your comment.

Our concept in this study is to express relationship between resectablity and survival at a single center. This is because in many papers, BR and UR-LA are collectively reported as locally advanced pancreatic cancer (LAPC), and UR-LA and UR-M are collectively reported as UR.

In our clinical practice, we felt that the conversion rates of BR and UR-PC were different, and the prognosis was also different. As mentioned in the discussion, there have been no paper that evaluated prognosis according to resectability and resection status in the present way, and we think the paper is original in this point.

According to your comment, we added the following phrases in Line 243-246

But there has been no study that comprehensively showed survival according to the initial resectability status or whether surgical resection was performed in a single institution. So, we think that it deserves a publication.

Comment 2:

Lines 54-56: “few studies have examined the conversion rates of BR and UR patients or the clinical outcomes of pancreatic cancer patients, including those who failed to undergo surgery”, references are laking. The sentence needs to cite references for these studies.

Thank you for your recommendation, we added four references in Line 56..

However, few studies have examined the conversion rates of BR and UR patients or the clinical outcomes of pancreatic cancer patients, including those who failed to undergo surgeryï¼»7-10ï¼½.

Comment 3:

The retrospective cohort includes 211 patients, which is considerable for PDAC. However, 43 patients (20% of the study population) have metastatic disease. It is very likely that the impact of surgery on survival of patients with systemic disease is very low if not null. Therefore, it is expected that their inclusion in analyses negatively affects survival estimations.

Thank you for your comments. To be honest, we didn’t think of this issue while writing this article, but it should be an important information to be described as a limitation of this study.

We added the following phrases in Line 234-237.

The retrospective cohort included 211 patients, which is considerable for PDAC. However, 43 patients, 20% of the study population, had metastatic disease. It is very likely that the impact of surgery on survival of patients with systemic disease is very low if not null. Therefore, it is expected that their inclusion in analyses negatively affects survival estimations.

Comment 4:

While termination of adjuvant chemotherapy and N+ status were confirmed as prognostic factors, surprisingly enough the presence of residual tumor after surgery failed to predict survival in multivariate analysis. Indeed, several studies have demonstrated that resection margins (R) status is an important independent predictor of survival: its lack of significance in Cox analysis rises to concerns about criteria used for patient selection.

Thank you for your advice, your comment is very meaningful for us. Hackert and Yamaguchi showed the significance of R0 or R0/1 resection. The favorable effect of adjuvant chemotherapy or the inclusion of UR-M patients as you suggested might have affected the results.

According to your comment, we added in Line 222-225.

Resection status, which other studies have demonstrated as an important independent predictor of survival, was not revealed as a prognostic factor in our study. This might be the influence of favorable effect of adjuvant chemotherapy or inclusion of patients with initially unresectable diseases.

Comment 5:

Lines 177-178 and 182-183, the following sentence appears repeated: “To complete the planned 6-months treatment, we need appropriate guidelines that are established for the proper management of adjuvant S-1 therapy.”

We would like to thank the Reviewer for pointing out our oversight. We have edited Line 177-178.

We would like to thank the Reviewer2 for his/her thoughtful and thorough review of our manuscript. Thanks to his/her comments and suggestions, we believe that our manuscript is now greatly improved

Round 2

Reviewer 1 Report

No changes to suggest

Author Response

Comment 1:

No changes to suggest

Response 1:

Thank you for your comment, and I am very happy to know your recommendation for publication.

Reviewer 2 Report

The authors addressed my concerns including the one on originality and the manuscript has been substantially improved

The following sentence at lines 245-246 needs to be deleted: "So, we think that it deserves a publication."

Clarity of sentences at lines 222-225 may be improved:  "Resection status, which other study has demonstrated as an important in dependent predictor of survival[8, 13], was not revealed as a prognostic factor in our study. This might be the influence of favorable effect of adjuvant chemotherapy or inclusion of patients with initially unresectable diseases.

Author Response

The following sentence at lines 245-246 needs to be deleted: "So, we think that it deserves a publication."

Clarity of sentences at lines 222-225 may be improved:  "Resection status, which other study has demonstrated as an important in dependent predictor of survival[8, 13], was not revealed as a prognostic factor in our study. This might be the influence of favorable effect of adjuvant chemotherapy or inclusion of patients with initially unresectable diseases.

Thank you for your comment.

According to your suggestion, I have deleted lines 245-246.

And I have revised the sentence according to the instructions

Resection status, which other study has demonstrated as an important independent predictor of survival[15, 23], was not revealed as a prognostic factor in our study. This might be the influence of favorable effect of adjuvant chemotherapy or inclusion of patients with initially unresectable diseases. And, UR patients with surgery are basically considered to be a group of patients who responded well to treatment with chemotherapy, and even if recurrence occurs, many of them might be survive for a long time due to good response of subsequent chemotherapy.